# In Silico Molecular Docking and Dynamics Simulation Analysis of Potential Histone Lysine Methyl Transferase Inhibitors for Managing β-Thalassemia

**DOI:** 10.3390/molecules28217266

**Published:** 2023-10-25

**Authors:** Yuvaraj Ravikumar, Pimpisid Koonyosying, Sirichai Srichairatanakool, Lakshmi Naryanan Ponpandian, Jayanthi Kumaravelu, Somdet Srichairatanakool

**Affiliations:** 1Department of Biochemistry, Faculty of Medicine, Chiang Mai University, Chiang Mai 50200, Thailand; ravikumaryuvi@gmail.com (Y.R.); pimpisid.s@cmu.ac.th (P.K.); 2Division of Hematology, Department of Internal Medicine, Chiang Mai University, Chiang Mai 50200, Thailand; rider_v.1@hotmail.com; 3Department of Herbal Pharmacology, Gachon University, Seongnam 13120, Republic of Korea; 4Department of Microbiology and Biotechnology, Bharath Institute of Higher Education and Research, Agharam Road Selaiyur, Chennai 600073, India

**Keywords:** β-thalassemia, fetal hemoglobin, EHMT2 inhibitor, structure-based docking, molecular dynamics

## Abstract

A decreased hemoglobin synthesis is contemplated as a pathological indication of β-thalassemia. Recent studies show that EPZ035544 from Epizyme could induce fetal hemoglobin (HbF) levels due to its proven capability to inhibit euchromatin histone lysine methyl transferase (EHMT2). Therefore, the development of EHMT2 inhibitors is considered promising in managing β-thalassemia. Our strategy to find novel compounds that are EHMT2 inhibitors relies on the virtual screening of ligands that have a structural similarity to *N*2-[4-methoxy-3-(2,3,4,7-tetrahydro-1*H*-azepin-5-yl) phenyl]-*N*4,6-dimethyl-pyrimidine-2,4-diamine (F80) using the PubChem database. In silico docking studies using Autodock Vina were employed to screen a library of 985 compounds and evaluate their binding ability with EHMT2. The selection of hit compounds was based on the docking score and mode of interaction with the protein. The top two ranked compounds were selected for further investigations, including pharmacokinetic properties analysis and molecular dynamics simulations (MDS). Based on the obtained docking score and interaction analysis, *N*-(4-methoxy-3-methylphenyl)-4,6-diphenylpyrimidin-2-amine (TP1) and 2-*N*-[4-methoxy-3-(5-methoxy-3*H*-indol-2-yl)phenyl]-4-*N*,6-dimethylpyrimidine-2,4-diamine (TP2) were found to be promising candidates, and TP1 exhibited better stability in the MDS study compared to TP2. In summary, our approach helps identify potential EHMT2 inhibitors, and further validation using in vitro and in vivo experiments could certainly enable this molecule to be used as a therapeutic drug in managing β-thalassemia disease.

## 1. Introduction

β-thalassemia is one of the most common inherited hemoglobinopathy disorders, where the homozygous or heterozygous forms result in a disproportional level of β-globin and α-globin synthesis, leading to feeble erythropoiesis and the reduced production of typical hemoglobin A [1,2,3]. A report suggests that around 1.5% of the global population are carriers of this genetic disease. In particular, this number is relatively high in Southeast Asia, where a carrier rate of almost 12.8% was recorded in Malaysia and 2.21% in China. According to the World Health Organization, over 40,000 infants are predicted to be diagnosed with β-thalassemia each year worldwide. In Thailand alone, 1 in 180 new births involve a β-thalassemia carrier, with a resulting minimum of 625 new cases, and a total of approximately 3250 new cases are reported yearly [1]. Patients with β-thalassemia often suffer from anemia, pain, and pulmonary hypotension symptoms, requiring constant medical attention [4,5]. The pathophysiologic differences in β-thalassemia in adults include the defective production of β-globin and functionally inactive red blood cells. Genetically, β-thalassemia is an autosomal recessive inherited disease where the homozygous or heterozygous forms result in a disproportional level of β-globin and α-globin synthesis, leading to feeble erythropoiesis and reduced production of typical hemoglobin A [6]. The existing treatment, such as blood transfusions and iron chelation therapy, helps manage iron overload but significantly affects specific vital organs, such as the liver, heart, and endocrine glands [7,8]. Furthermore, constant blood transfusion treatment might not be ideal for β-thalassemia-involved patients. Therefore, the search for the perfect alternative for managing abnormal erythropoiesis is of the utmost importance, and recently, much attention has been devoted to modulating the fetal hemoglobin (HBF) levels by inducing or reactivating the γ-globin expression through epigenetic regulators [9,10].

The oxygen-carrying hemoglobin, the heterotetramer in adults, comprises two α- and β-globin chains. However, the erythrocytes preferentially chose the γ-globin over the β-globin to form a tetrameric HbF during fetal development Figure 1 [11]. After childbirth, followed by several months, the transition of the γ-globin with the β-globin occurs by a mechanism called globin switching [12,13]. This suggests that higher HbF synthesis in RBCs could alleviate the progression of β-thalassemia. In such accordance, the therapeutics that could enhance HbF production have gained significant attention over the last few years, and reports that demonstrate the role of epigenetic regulators in HbF production have been well-documented [14,15]. Epigenetic regulators such as histone methyltransferases, which catalyze histone arginines and lysines with the help of *S*-adenosyl-methionine (SAM), play a vital role in DNA damage repairs, meiosis, germ cells, and embryonic development processes. Studies show that inhibiting the G9a, also known as EHMT2 (euchromatic histone lysine methyl transferase 2) activity, can induce fetal hemoglobin (HbF) production in human adult erythroid cells [16,17,18]. Various classes of small molecule G9a inhibitors have been identified, and their pharmacokinetic properties are well investigated. For instance, UNC0638, which contains a lysine-like structure, a 3(pyrrolidine-1-yl) propoxy group, UNC0642, and BIX01294 have all shown promising inhibition potential against G9a. These quinazoline compounds act as suitable inhibitors; the poor membrane permeability makes them unsuitable for use as a drug clinically [1]. More recently, a non-quinazoline-based drug identified as EPZ035544 from Epizyme was reported at a conference. When evaluated in an in vitro assay, this compound showed an IC_50_ value of 16.9 nm against G9a. However, like the earlier reported compounds, this, too, showed poor oral availability [1]. Owing to such poor pharmacokinetic profiles of the existing compounds, there is a demand for novel compounds with lower toxic effects and good pharmacokinetic properties.

In the current decade, the identification of new drug compounds has benefitted from the advent of computer-aided drug discovery methodologies and molecular docking studies. Virtual screening (VS) has allowed researchers to rapidly analyze many compounds from specific databases to find promising hit molecules [19,20]. Meanwhile, molecular docking has been used successfully to determine and understand the affinity and stability of the interactions that arise from ligand–protein complexes. The methods are expeditious and prevent final step drug failure as the number of compounds for biological evaluation is confined to small numbers [21]. In addition, as the biomolecules within the body are in constant motion, it is a prerequisite to understand the dynamics of the target protein and the docked compound before entering clinical studies [22]. Owing to such high significance, the applications of these computational tools for drug discovery against various diseases have been increasing daily. However, only a few reports have been documented concerning β-thalassemia conditions [1]. Therefore, in this study, by utilizing the previously reported Compound 13 (*N*2-[4-methoxy-3-(2,3,4,7-tetrahydro-1*H*-azepin-5-yl) phenyl]-*N*4,6-dimethyl-pyrimidine-2,4-diamine, hereafter called as F80), which acts an inhibitor against G9a, a detailed virtual screening was performed in the PubChem database [23,24]. The compounds with a structure similarity of over 90% with F80 were chosen. The obtained libraries were then subjected to molecular docking and dynamic simulations to identify the potential compounds that could be used as drug molecules for treating β-thalassemia. 

## 2. Results

In the current decade, the virtual screening of compounds has become a looming trend. It has been confirmed to be an assuring method for discovering potent inhibitors that can be used against various diseases. Being economical and rapid, its application for developing promising drug candidates for regulating critical enzymes involved in epigenetic regulation is burgeoning and prompt. Inhibition of epigenetic regulating enzymes as a treatment against genetic disorders has been endorsed for sickle cell disease. This makes the EHMT2-SET domain an exciting drug target, especially as its crystal structure has been well-documented. Figure 2 depicts the schematic work plan of this study employed for attaining hit discovery against EHMT2. Thus, to identify novel inhibitors for EHMT2, PubChem, which comprises around 115 million compounds, was chosen, and a virtual database screening strategy was employed. 

### 2.1. Virtual Screening and Molecular Docking

The reported crystal structure of the EHMT2 (PDB ID 7BUC) complexed with F80 was selected, and the compound F80 was further used to screen a compound library in the PubChem database. From the search, we aimed to select only compounds with >90% structure similarity with the F80. Around 985 compounds were retrieved and docked into the EHMT2 binding pocket. After docking, the compounds that showed better docking scores, along with F80 and *N*~2~{4-Methoxy-3-[3-(pyrrolidin-1-yl)propoxy]phenyl}-*N*~4~,6-Dimethylpyrimidine-2,4-diamine or Compound 5, hereafter called as N47, was selected, and the compounds were ranked based on the docking score. Further, only the TOP10 compounds were chosen to increase the potential hits, and their binding poses were analyzed further. Legitimate binding poses with EHMT2 were detected in all the TOP10 compounds. Accordingly, the compounds in the top 10 ranking based on the docking score were identified. Table 1 and Appendix A show the total 10 selected compounds’ docking scores, the type of interaction with the active site residues, and the calculated bond distances. The docking scores were determined using Autodock Vina, and the range of docking scores lay between −7.5 and −10.7 kcal/mol. Compound *N*-(4-methoxy-3-methylphenyl)-4,6-diphenylpyrimidin-2-amine (TP1), and 2-*N*-[4-methoxy-3-(5-methoxy-3*H*-indol-2-yl)phenyl]-4-*N*,6-dimethylpyrimidine-2,4-diamine (TP2) exhibited good docking scores of −10.7 kcal/mol, and −10.3 kcal/mol respectively when compared to F80 and N47. Both hydrogen bonding and hydrophobic interactions with the protein mediated the binding of the inhibitors. All F80, N47, TP1, and TP2 had hydrophobic or hydrogen bonding with the protein. Our putative drug targets, TP1 and TP2, had hydrogen bond interactions. A maximum number of hydrogen bonds were formed in TP2, forming three hydrogen bonds with EHMT2, while TP1 formed a single hydrogen bond. The amino acids that were interacting with F80 and TP2 were Cys1098, Leu1086, Asp1083, Phe1158, Ser1084, Asp1088, and Arg1157, and interestingly, both these compounds commonly formed a H-bond interaction with Asp1083, Ser 1084, and Asp1088, respectively. The binding pose, or the specific orientation of the compound about the protein, is depicted in Figure 3. Additionally, the molecular interactions from TP3 to TP10 with EHMT2 are shown in Appendix A. With these initial results, the compound’s toxicity and pharmacokinetic profile were further analyzed.

### 2.2. Physiochemical and Drug-Likeness Analysis

Besides its efficiency, any drug should be able to pass through a wide range of barriers to elicit the drug response at the target site of action. Recently, the availability of in silico methods for rapidly predicting pharmacokinetic and toxicity properties, including absorption, distribution, metabolism, and excretion (ADME) of novel identified compounds in the human body, has curtailed the time and tedious experimental methods. Therefore, the SwissADME and ADMET 2.0 software was utilized to determine the critical pharmacokinetic properties of F80, N47, and the top two compounds that showed higher docking scores, TP1 and TP2, to proceed with the next step of the proposed workflow. First, the oral bioavailability of the selected compounds was assessed using Lipinski’s rule of five, and we found that all the compounds passed the drug-likeness and exhibited good oral bioavailability. The other vital parameters of medicinal chemistry, such as compound synthetic accessibility, Sp3, and the golden triangle, were predicted. Results showed that N47, TP1, and TP2 had satisfied all the parameters, while F80 failed to pass the Sp3 property. In absorption criteria, all the tested compounds had met the gastrointestinal (GI) absorption P-glycoprotein uptake and Madin–Darby canine kidney (MDCK) permeability, which denoted that the identified compounds could be well absorbed in the intestine and taken up by the P-glycoproteins (P-gp) (Appendix A). In addition, the compounds were also screened for various drug-likeness properties. None of the compounds failed to pass the Lipinski (Pfixer), Ghose, Veber (GSK), and Egan (Pharmacia) filters. Along with these, the tested compounds were also able to succeed in the pan-assay interference compound (PAINS) parameter. The results of the pharmacokinetic properties are summarized in Table 2.

### 2.3. Molecular Dynamics Simulations

Implementing molecular dynamic simulations (MDS) aids the precise analysis of the dynamics of biological macromolecules set under a controlled physiological environment. MDS is a computational method used for analyzing the physical interactions in a biophysical system in which docked complexes’ flexibility and structural differences can be observed during the simulation time. As the principal aim of this MDS study was to establish the vital intermolecular interactions of the bound ligands and binding stability with the EHMT2 active site, the simulation was performed on the crystal structure of the EHMT2 SET domain unbound and protein–ligand complexes. (i) Protein alone [7BUC-APO], (ii) Protein-bound to crystal ligand F80 [7BUC-F80], (iii) Protein-bound to known inhibitor N47 [7BUC-N47], (iv) Protein-bound to top compound selected from docking *N*-(4-methoxy-3-methylphenyl)-4,6-diphenylpyrimidin-2-amine [7BUC-TP1], (iv) Protein-bound to second top compound selected from docking 2-*N*-[4-methoxy-3-(5-methoxy-3*H*-indol-2-yl)phenyl]-4-*N*,6-dimethylpyrimidine-2,4-diamine [7BUC-TP2]. By performing MDS, we could deduce the key information about the dynamic action of tested compounds by evaluating the trajectories performed at 200 ns time function within the solvated medium. By maintaining the physiological conditions, the changes that occurred in protein-unbound and protein-bound ligand complexes were analyzed based on various parameters such as root mean square deviations (RMSDs) for all the ligands and backbone atoms, the root mean square fluctuations (RMSFs) for the individual amino acids, gyration radius, and intermolecular hydrogen bond formation.

#### 2.3.1. RMSD

The subtle changes in the overall structure and conformational stability of all the selected docked complexes were assessed by evaluating the protein backbone atoms RMSD against simulation time. As seen in Figure 4, with reference to the RMSD from the crystal structure, in the 7BUC-APO form, the RMSD values increased sharply for 152 ns, reached 0.7 nm, and attained equilibrium in the range of 0.7 to 0.6 nm throughout the 200 ns. In the 7BUC-F80 complex, the RMSD initially fluctuated up to 155 ns and remained stable throughout the simulation. The range of measured RMSD from 155 ns to 200 ns was 0.6 nm. In the 7BUC-N47 complex, the complex gained stability much earlier than F80, where RMSD fluctuations stopped at 130 ns and remained in equilibrium until 200 ns. The RMSD values recorded at 130 ns were 0.6 nm, and at 200 ns, it was 0.5 nm. Followingly, the RMSD values for TP1 and TP2 were measured. Minimal fluctuation was noted in the TP1-7BUC complex, where the complex attained stability very shortly, 0.6 nm in 15 ns, and no significant deviation was observed further. Unlike TP1, in the TP2-7BUC complex, the stability in RMSD values was seen only after 50 ns and remained stable until 180 ns. The values were in the range between 0.5 to 0.7 nm. Slight perturbations were present in the TP2-7BUC complex.

#### 2.3.2. RMSF

RMSF is defined as the displacement of a particular atom or group of atoms relative to the reference structure and averaged over the number of atoms. This parameter is notably effective for investigating the individual residue flexibility present in the protein backbone. Figure 5 demonstrates the RMSF analysis of 7BUC-APO, 7BUC-F80, 7BUC-N47, 7BUC-TP1, and 7BUC-TP2 complexes. In addition, average RMSF values are calculated against the simulation timescale of 0 to 200 ns. The EHMT2 SET domain average RMSFs from 0 to 200 ns for 7BUC-APO, 7BUC-F80, 7BUC-N47, 7BUC-TP1, and 7BUC-TP2 protein complex proteins were 0.09 ± 0.04 nm, 0.09 ± 0.07 nm, 0.10 ± 0.05 nm, 0.07 ± 0.06 nm respectively. Based on the RMSF values, albeit no significant differences in values were observed in each system, fewer fluctuations were noted in the residues present in the β-sheet regions of each complex. Apparent increases in RMSF values, particularly in the residues between 60 and 80, were found in TP1 (Appendix A). In addition, residues present from 160 to 200 were identified as ligand-interacting residue regions, and significant differences in RMSF values between TP1 and TP2 were observed (Appendix A). Ideally, higher RMSF fluctuations were seen in TP1, and lower fluctuations were noted in TP2, and the RMSF values of TP2 in these regions were similar to those of F80, N47, and apoenzyme.

#### 2.3.3. Radius of Gyration

Like RMSD and RMSF, MDS radius of gyration (Rg) is another crucial factor in evaluating a protein’s structural integrity and compactness, eventually indicating the system’s stability. The Rg can be described as the mass-weighted root mean square distance of atoms from their center of mass. The competence shape folding of the overall structure at different time points during the trajectory can be seen in the Rg plot. In such accordance, time-dependent Rg analysis for the simulated complexes is depicted in Figure 6. The average Rg value from 0 to 200 ns for 7BUC-APO, 7BUC-F80, 7BUC-N47, 7BUC-TP1, and 7BUC-TP2 protein complex proteins were 2.35 ± 0.01 nm, 2.34 ± 0.01, 2.36 ± 0.01 nm, 2.34 ± 0.01 nm respectively. Significant anomalies in compactness were noted in 7BUC-F80 and 7BUC-TP2 complexes. As shown in Figure 6, a sudden spike in the Rg values was noted in F80-7BUC after 50 ns and maintained higher values throughout the trajectory. In the first 50 ns, Rg values for the TP1-bound complex decreased (2.0–1.90 nm) and remained constant. In TP2-bound protein, the Rg value initially increased up to 50 ns, where the maximum Rg value of 2.10 was attained within 25 ns simulation and then reduced to 2.0 and increased above 2.0 at the end of trajectory. Albeit slight changes in the Rg values of the protein when complexed with TP1 and TP2 were noted, no significant difference between the shape folding of the complex at different time points during the trajectory was observed. This suggests that the protein complex is structurally stable throughout the simulation.

#### 2.3.4. SASA Analysis

Solvent-accessible surface area (SASA) refers to the protein area well exposed to interact with the nearby solvent molecules. It has become a valuable probe for understanding protein folding and stability and measuring the compactness of the protein’s hydrophobic core. Thus, the change in SASA was analyzed (Figure 7). The EHMT2 SET domain average SASA value from 0 to 100 ns for 7BUC-APO, 7BUC-F80, 7BUC-N47, 7BUC-TP1, and 7BUC-TP2 protein complex proteins were 202.04 ± 3.14 nm, 203.5 ± 3.57, 208.6 ± 6.16 nm, 203.1 ± 4.88 nm respectively. None of the complexes seems to have significant deviation except for TP2. In a detailed analysis, the continuous variation is noted in TP2, where the decrease in the area was initially observed until 100 ns and remained stable, followed by an increase in the area at the end of simulation time. Thus, the values can be attributed as clear evidence of minimal change in the conformational states of the target protein. The fluctuations pattern of SASA values in TP2 was similar to that of the pattern of Rg documented. More importantly, only slight variations were observed when comparing the apo form with all the ligand-bound complexes.

#### 2.3.5. Molecular Interactions Analysis

In general, protein–ligand complexes are stabilized by the formation of hydrogen bonds. To demonstrate the stability of EHMT2-ligand complexes, the hydrogen bond formation during MDS was monitored. A meticulous investigation of hydrogen bonds between each compound and the EHMT2 SET domain was predicted under the influence of the CHAARMM force field. The EHMT2 SET domain H-Bond result of the complex with 7BUC-F80, 7BUC-N47, 7BUC-TP1, and 7BUC-TP2 is depicted in Figure 8. The H-bond occupancy analysis for all the complexes at 200 ns shows us that the highest number of H-bond formations was found in 7BUC-F80 (100), followed by 7BUC-N47 (85), and the lowest was recorded in 7BUC-TP1 complex (65). The number of H-bonds slightly increased in the 7BUC-TP2 (75) complex. The intra-molecular H-bond analysis shows that the maximum number of H-bonds was formed in the N47-7BUC and TP2-7BUC complex. During the trajectory analysis, the maximum number of H-bonds observed in the N47-7BUC complex is five. Next to N47, the higher number of H-bonds interacted with the TP2-7BUC complex is four. In contrast, in the TP1-7BUC complex, a minimal number of H-bonds is formed where the number of H-bonds does not exceed two during the entire simulation period. In addition to the number of H-bonds, the residue of 7BUC that results in H-bond interaction with each ligand was envisaged. The analysis found that the hydrogen bonds formed and their interactions were the same as the docking complexes. Leu1086, Asp1083, Asp1078, and Arg1157 are the amino acids in H-bond formation in docking and MDS. In addition to hydrogen bonding, hydrophobic interactions also helped in binding ligands with the EHMT2 active site. As seen in Appendix A, the key amino acids that form hydrophobic contacts with F80 are Ile1136. In the N47 bound complex, Asp1074 and Phe1088 give rise to hydrophobic contacts. Many more hydrophobic interactions were found in the TP1 and TP2 bound complexes. The other amino acids that play a vital role in ligand binding, such as Arg1080, Val1094, Asp1084, Phe1158, Phe1087, and Leu1086, help establish hydrophobic interactions with our putative drug targets. Further stabilization of these ligands in EHMT2 occasionally was facilitated by π-π stacking cation- π interactions. For example, with F80, Arg1123 established cation- π interaction. Similarly, Phe1087 forms a by π-π stacking interaction when N47 is bound with EHMT2. Interestingly, both π-π stacking and cation- π interactions were found to be formed while analyzing the TP1-7BUC complex MDS trajectories. Phe1166 aids in forming π-π stacking within a distance of 5.7 Å. Arg1157 results in forming cation- π interactions within a distance of 4.2Å. Overall, from the molecular interaction analysis derived from MDS trajectories of each ligand-bound complex, these interacting amino acids might be determined to play a crucial role in binding with the ligands of the EHMT2 active site.

#### 2.3.6. Principal Component Analysis

Principal component analysis (PCA) is a statistical technique that can reduce the dimensionality of data and extract important features or patterns. It is a standard tool used in structural biology to analyze the conformational changes in proteins or protein–ligand complexes. Therefore, we performed the PCA analysis on the EHMT2 SET domain H-Bond result of the complex with 7BUC-APO, 7BUC-F80, 7BUC-N47, 7BUC-TP1, and 7BUC-TP2. The conformational changes in EHMT2 attributed to the ligands binding and the cumulative motions of the MD trajectories were analyzed, and the proteins were conformational in examining the structural differences between the various protein samples. The covariance matrix was calculated after removing translational and rotational motion from the data. The covariance matrices of the backbone C alpha atoms were then diagonalized to obtain the eigenvectors and eigenvalues. The first two projections (PC 1 and PC 2) with the highest eigenvalues were considered the collective motions of the C alpha backbone atoms. The 2D plots of PC1 and PC2 for the different protein samples are shown in Figure 9. The PCA analysis showed that the F80, N47, TP1, and TP2 complexes are more rigid than the Apo protein. This is evident from the smaller area covered by the plots for these complexes and the fact that they are more closely clustered. Interestingly, unlike other complexes, the 7BUC-TP1 complex exhibits strikingly different conformations and is not inverted. This suggests that TP1 binding with EHMT2 has some changes in conformations from inactive to active structure. These dynamic changes would explain the higher RMSF observed in 7BUC-TP1.

#### 2.3.7. MM-PBSA Analysis

Albeit the docking procedure and its related scoring value establish excellent assuming power in identifying the best ligand pose binding with the protein active site, they are undecisive to confirm the compound’s rank order with correlated to binding affinities. Such unreliability could be due to relentless assumptions operated by scoring functions, which can considerably lower the accuracy in calculating the docking scores. To circumvent such issues, lately, the utilization of physical energies such as surface accessibility area and solvation energy by molecular mechanical force field cater ligand binding energy with high aggregable certainty. Hence, for each ligand, the best-docked pose obtained from earlier docking studies was subsequently employed for MM-PBSA post-docking analysis. MM-PBSA results in minute differences in ligand conformations in the protein active site, and these changes, in turn, result in the stabilization of the protein–ligand complex. Owing to such high significance, we examined the relative binding strength of 7BUC-F80, 7BUC-N47, 7BUC-TP1, and 7BUC-TP2 complex via the MM-PBSA approach. The calculated binding energy along with Van der Waals, electrostatic, and polar solvation energy attained during stable simulation trajectory are summarized in Table 3.

The table presents the results of energy calculations for four different systems: 7BUC-F80, 7BUC-N47, 7BUC-TP1, and 7BUC-TP2. Each row represents a specific system, while the columns represent various energy components: Van der Waals energy, electrostatic energy, polar solvation energy, and binding energy. These energy values are reported with their corresponding uncertainties in kilojoules per mole. The first system, 7BUC-F80. The Van der Waals energy is calculated to be −171 ± 10 kJ/mol, indicating an attractive interaction between the molecules. The electrostatic energy is slightly negative at −0.1 ± 5 kJ/mol, suggesting a repulsive interaction. The polar solvation energy is positive, with a value of 698 ± 12 kJ/mol, indicating that the system gains energy when immersed in a polar solvent. Lastly, the binding energy is −119 ± 11 kJ/mol, suggesting that the system is stable and releases energy upon binding. Moving on to the second system, 7BUC-N47, we observe similar patterns. The Van der Waals energy is −224 ± 16 kJ/mol, indicating attractive molecule interactions. The electrostatic energy is more negative at −28 ± 12 kJ/mol, suggesting a stronger repulsive interaction than the first system. The polar solvation energy is positive, with a value of 149 ± 35 kJ/mol, indicating an energy gain in the presence of a polar solvent. The binding energy is −126 ± 22 kJ/mol, showing a stable system that releases energy upon binding. Moving on to the third system, 7BUC-TP1, we observe similar trends. The Van der Waals energy is −230 ± 9 kJ/mol, suggesting attractive interactions. The electrostatic energy is −26 ± 7 kJ/mol, indicating repulsive interactions. The polar solvation energy is positive, with a value of 138 ± 22 kJ/mol, meaning an energy gain in the presence of a polar solvent. The binding energy is −141 ± 17 kJ/mol, suggesting a stable system that releases energy upon binding. Lastly, for the fourth system, 7BUC-TP2, we observe similar trends as well. The Van der Waals energy is −178 ± 14 kJ/mol, indicating attractive interactions. The electrostatic energy is significantly more negative at −544 ± 23 kJ/mol, suggesting strong repulsive interactions. The polar solvation energy is positive, with a value of 276 ± 24 kJ/mol, indicating an energy gain in the presence of a polar solvent. The binding energy is −465 ± 24 kJ/mol, suggesting a stable system that releases significant energy upon binding. The binding energy values range from around −470 kJ/mol to −120 kJ/mol. These values are negative, indicating stable systems that release energy upon binding. The electrostatic, polar solvation and binding energy value of 7BUC-TP2 varies significantly with other complexes. Despite the difference in energy values, we could not deduce or reason the difference shown in the MM-PBSA analysis. The magnitude of the binding energy reflects the strength of interactions between the binding partners. More negative values suggest stronger binding and higher stability of the complex. The variations in binding energy among the systems can be attributed to differences in molecular structures, complementarity, and the nature of intermolecular interactions.

## 3. Discussion

One of the main aims of computational-based bioinformatics is to find the potential drug candidates that will boost the effectiveness, efficiency, and shortness of time for drug development. In such accordance, considering the minimal toxicity, good compatibility, and high oral bioavailability of the amine derivatives that were earlier reported, we, in this study, utilized F80 as a starting compound and virtually screened the structurally similar compounds in the PubChem database. Further, to identify the potentially hit compounds, the screened compounds are narrowed down using a molecular docking approach and simulation studies.

This study utilized the PubChem database, which caters to an extensive library of ligands helpful for 3D molecular docking. Around 985 ligands having 95% structural similarity with F80 compounds were chosen, and molecular docking was performed against 7BUC. The 7BUC that encodes for the EHMT2-SET domain had an ortho-steric binding site which is packed with amino acids, namely, Cys1098, Leu1086, Asp1083, Phe1158, Ser1084, Asp1078, and Asp1088. These amino acids were found to have interactions with the co-crystallized compound F80. Initially, we attempted redocking two agonists, F80 and N47, with 7BUC. The redocking results revealed that the ligand F80 is well-docked with the active site of the EHMT2-SET domain, where the interactions of the docked F80 complex were similar to the co-crystallized one. This is evident with the hydrogen bond formed between F80 and Leu1086. In addition, the N47, which was docked with 7BUC, showed the ligand was well bound in the active site vicinity, which was further confirmed by the type of interactions formed with the enzyme’s amino acids. The initial findings attained during re-docking analysis confirm that docking studies performed using Autodock are highly reliable and, in turn, enticed us to do docking for all 985 ligands that were retrieved from the PubChem database. The ligands exhibiting docking scores higher than F80 and N47 were filtered out among the docked compounds. The docking scores for the selected ligands, as shown in Table 1, were between −10.7 and −9.8. Compared to F80, the compounds TP1 and TP2 showed higher docking scores of −10.7 and −10.3, respectively. The docking studies analysis revealed that similar conformations for TP1, TP2, and F80 were obtained. Albeit multiple docking poses with the TP1 and TP2 were obtained, the pose with maximum interactions and hydrogen bonding was taken. As shown in Appendix A in F80, at 200 ns time, the oxygen atom of Leu1086 was involved in the hydrogen bonding with the nitrogen atom of the ligand. Similar hydrogen bonding interactions were also present in TP1 and TP2. In TP1, the oxygen atom present in Asp1086 forms a hydrogen bond with the nitrogen atom of the ligand, while in TP2, the oxygen atom of the ligand includes a hydrogen bond interaction with the NE atom of Arg1157. It is noteworthy that hydrogen bond interactions between the ligands and 7BUC occurred with different amino acids, and the distance between F80 and TP1 is 2.92 and 2.88, while 3.29 Å with TP2, respectively. The hydrogen bond formation is crucial as it aids in better ligand–protein interactions primarily by removing the protein-bound water molecules. Our findings were in accordance with the earlier studies, where a study reported by Katayama et al. showed that the amino indole derivative DS79932728 has the same binding site, and the amino acids that participated in hydrogen bonding were Leu1086 [24]. It is also noteworthy that the molecular interactions of all top 10 ranked compounds docked with EHMT2 were analyzed, and we found out that TP3, TP4, and TP5 do not form any conventional hydrogen bond (Appendix A). Only TP1, TP2, TP6, TP9, and TP10 showed conventional hydrogen bond interactions, and based on the docking score and maximum hydrogen bond and hydrophobic interactions present, we selected only TP1 and TP2 for further MDS analysis.

The MDS study produced satisfactory results in RMSD and RMSF for all the tested compounds. No significant instability was noted between F80-7BUC, N47-7BUC, TP1-7BUC, and TP2-7BUC. All the complexes tended to remain stable over a 200 ns simulation time. Further, comparing the RMSF versus residue number of TP1 and TP2 revealed an area that seems noticeably less flexible in TP2 from residues 1085–1098. The amino acids in these regions form different interactions with TP1 and TP2. In general, reports suggest that the lower RMSF values help improve the structural rigidity of the ligand-bound complexes [25,26]. From the RMSF analysis, it is evident that TP2 tends to be much more rigid than TP1. Nevertheless, pronounced fluctuations were not observed, and most of the residues showed RMSF values equivalent to F80 and N47, which indicates the maintenance of well-structured regions during and after complex formation. Next, we measured the binding energy value for the selected compounds by correlating the binding of F80, TP1, and TP2 with 7BUC using simulated trajectory-based MM-PBSA outputs. The calculated binding energies, as depicted in Table 2, were −119 ± 11, −126 ± 22, −141 ± 17, and −465 ± 24 kJ/mol. Based on the binding free energy values, it is assumed that the biological activity of TP1 and TP2 will be much higher than the other compounds. Also, by further exploring and understanding which energy parameters favor binding each compound to 7BUC, the investigation of individual energy components such as Van der Waals, electrostatic, and polar solvation energy is of utmost importance. These energy values arise from the interactions developed due to the contributions of residues, backbones, and side chains. The maximum contribution of binding affinity of the TP1 and TP2 bound protein complex was from Van der Waals energy. Likewise, the electrostatic energy influences TP1 and TP2 binding with the EHMT2. Taken together from the MM-PBSA analysis, it is assumed that Van der Waals/electrostatic/polar solvation energy plays a vital role in contributing to the TP1/TP2 binding. In addition to the aforementioned energies, hydrogen bond interactions for TP1, TP2, and F80 binding with EHMT2 are essential, as it is well known that a hydrogen bond remains a crucial factor for maintaining the stability of ligand-bound complexes. In such accordance, while analyzing the docking results, we found that F80 forms an H-bond with Leu1086. This was confirmed with MDS trajectories, where the same Leu1086 of 7BUC includes an H-bond with F80 during a complex formation. Similar results were obtained in N47, TP1, and TP2. Likewise, in both N47 docking and MDS studies, Arg at a 1157 position exhibits an H-bond with N47. This shows that our docking results were reliable and consistent in the TP2-7BUC complex. Here, H-bonding between TP2 and proteins in both docking and trajectories was established by Leu1086. Surprisingly, in the TP1-7BUC complex, the residues forming the H-bond during docking and trajectories obtained in MDS differed. Arg1157 was found to create an H-bond in the docked complex; in contrast, while analyzing trajectories, Arg1157 did not seem to have any interaction, and instead, Asp1078 had an interaction with TP1 via the H-bond. Nevertheless, Asp1078, present in the 7BUC active site, is also presumably considered to play a key role in forming interactions with the ligand. Furthermore, the superimposition of respective ligand-bound docked complexes with MDS complexes reveals that the overall structure of 7BUC is well aligned, and no significant movement in each ligand and H-bond forming residues were observed. Overall, this study found two potential novel EHMT2-SET domain inhibitors based on the backbone of the F80 structure. Our molecular docking and MDS analysis reveal that molecular interaction differences between residues were observed for TP1 and TP2 while binding with EHMT2. We believe both these compounds have strong potential for use as a drug molecule in managing β-thalassemia. In particular, TP1 can be a potential hit candidate as we could also find the formation of non-covalent interactions such as π-π and cation-π interactions with EHMT2. Generally, cation-π interactions are highly important to the function and pharmacology of various ion channel receptors and membrane proteins. For instance, binding nicotine with Ach receptors in the brain necessitates cation-π interaction [26,27,28]. In such high significance, we could find Phe1166 forming π-π stacking and Arg1157 forming cation- π interactions with TP1. This shows that these interactions make the F80-bound EHMT2 more stable and could inhibit its activity completely. However, future studies involving in vitro methods are essential to authenticate the novel compounds identified through this study.

## 4. Conclusions

This study employed a comprehensive virtual screening approach involving a structure based on successfully identifying new compounds that are EHMT2 inhibitors. To enact this goal, computational screening of a library of 985 compounds obtained through a similarity search of earlier reported EHMT2 inhibitors was subjected to docking studies. The identified hits with the highest binding affinity were ranked based on the auto dock score lower than 15 kcal/mol. The top two ranked compounds fulfilled the prerequisite ADMET properties and cleared the false-positive evaluation in the PAINS analysis. In addition, the docking simulations combined with an MM-PBSA analysis showed that the selected compounds TP1 and TP2 had higher binding affinities when compared to the reported F80 and N47. Subsequently, the best-docked pose for F80, N47, TP1, and TP2, when complexed with the 7BUC active site, was subjected to 200 ns simulations to reveal the stability of the molecular interactions and the binding strength and identify novel interactions that were absent in the docked complex. While comparing the RMSD, RMSF, and Rg analysis and arrangement of intermolecular interactions of all ligand complexes observed in MDS time, TP1 exhibited the best binding mode and stability profiles when bound with the EHMT2 active site. The IFD and MDS analysis together defined the significance of ideal hydrogen bonds and hydrophobic interactions with vital residues of EHMT2 active sites such as Arg1080, Asp1078, Asp1088, Arg1157, Val1094, Phe1158, and Phe1166. In summary, the TP1 compound showed promising potential and can be further developed as a new EHMT2 inhibitor when coupled with experimental results in vitro.

## 5. Materials and Methods

### 5.1. Virtual Screening

The virtual screening was performed using the PubChem database, which contains over 96 billion chemical structures. This free online database was utilized in this study to identify the potential molecules that could be EHMT2 inhibitors [29]. For this objective, a similarity search was employed to select the compounds that have maximum similarity with the compound *N*2-[4-methoxy-3-(2,3,4,7-tetrahydro-1*H*-azepin-5-yl) phenyl]-*N*4,6-dimethyl-pyrimidine-2,4-diamine. The PubChem database, which implements the Tanimoto-based 2D fingerprint similarity search method, results in generating a library comprising 985 hits with >90% similarity. The structures of these compounds were saved in Structure-Data File (SDF) format and were subsequently used for further analysis.

### 5.2. Protein Preparation

The crystal structure of the human EHMT2 SET domain complexed with inhibitor F80 (PDB ID: 7BUC) was recovered from the RCSB Protein Data Bank [23], and protein preparation was performed in Biovia Discovery Studio [30]. This allows us to process the protein suitable for further docking procedures. First, the compounds F80 and N47 that were co-crystallized with the EHMT2 SET domain were removed manually, followed by removing water molecules and other heteroatoms except SAM and Zn^2+^ ions. Next, hydrogen atoms were added, assigning bond order and filling missing atoms and residues if present. Further, His, Asp, and Glu residue’s protonation state were determined for optimizing the hydrogen bond network.

### 5.3. Molecular Docking Analysis

In order to restrict the number of promising hits and assure the accuracy of the virtual screening, all 985 compounds retrieved from the PubChem database were further subjected to docking analysis. The molecular docking studies were performed using AutoDock Vina [31]. First, the 3D structures of the ligands were obtained from PubChem and prepared for docking using the Open Babel software 3-1-1 [25,26]. The receptor crystal structure of the EHMT2 SET domain in a complex with compound 13 was also prepared for docking. The docking protocol was set up in AutoDock Vina, including setting the search space and grid box dimensions of 60 × 60 × 60, and the docking runs were initiated for triplicate docking. The position of SAM, catalytic Zn^2+^ ion, and the ligand binding region in the crystallized EHMT2 SET domain was analyzed. The grid box defining the active site was generated. More importantly, as our target enzyme is a metal-dependent enzyme, the metal constraints during receptor grid preparation were considered as they might result in metal-coordination binds during docking studies. After the successful completion of docking, the resulting docking poses were analyzed for their binding affinity, and the top poses were selected for further analysis. Further, the compounds were sorted based on the affinity kcal/mol values. The docking scores for F80 and N47 were taken as reference, and the compounds that have score values closer to them were treated as filtering criteria. This results in identifying 10 top-ranked compounds as potential EHMT2 SET domain inhibitors with docking energy ranging from −7.5 to 10.7 kcal/mol. In addition, the best binding poses, modes, and interactions of top-ranked compounds were analyzed further using Pymol and Discovery studio visualizer [30,32].

### 5.4. Physiochemical and Drug-Likeness Properties 

The physicochemical properties of F80, N47, TP1, and TP2 were analyzed using Swiss-ADME software (http://www.swissadme.ch/, accessed on 1 April 2023). The drug-likeness properties such as molecular weight, H-bod donors, H-bond acceptors, logP Values, and other important criteria are assessed as per Lipinski’s rule of five (http://www.swissadme.ch/, accessed on 1 April 2023). It is believed that the compounds that obey Lipinski’s rule of five have better polarity and better folding and tend to exhibit good therapeutic effects. Followingly, ADMET 2.0 was used to determine parameters such as toxicity, metabolism using cytochrome P450, gastrointestinal absorption, and excretion (https://admetmesh.scbdd.com/, accessed on 2 April 2023).

### 5.5. Molecular Dynamics

Molecular dynamics (MD) simulation is a computational technique that uses Newton’s laws of motion to study the movement of atoms in a molecule. The top 2 ranked compounds based on the docking energy along with F80 and N47 were subsequently employed for MD simulation studies for assessing the binding mode of each compound with the active site of the EHMT2 SET domain and also to demonstrate the interactions of ligand–protein in more detail fashion. The simulation was performed using the Gromacs software package-2019.4, a widely used and well-established MD simulation software (http://www.gromacs.org/, accessed on 9 May 2023). CHARMM-GUI force field was used for the MDS studies. The coordinates of the ligands used for semiempirical calculation for this study are X-axis 5.894, Y-axis 20.361, and Z-axis 26.496, respectively. The first step in the simulation process was minimizing the protein–ligand complex in a vacuum. This was performed using the steepest descent algorithm, which involves iteratively adjusting the complex’s atomic coordinates to reduce the system’s potential energy. After the minimization, the complex was solvated in a periodic water box using the SPC water model. The SPC water model is a simple model representing water molecules as a single-point charge and is often used as a starting point for more complex water models. The complex was also maintained at a salt concentration of 0.15 M by adding appropriate sodium and chloride ions. The resulting complex was then subjected to an NPT (constant pressure, constant temperature) equilibration phase, followed by a production run for 200 ns (nanoseconds) in the NPT ensemble. The NPT ensemble simulates systems at constant temperature and pressure, commonly encountered in biological systems. Finally, the trajectory of the simulation was analyzed using various tools provided by the Gromacs software package -2019.4, including the protein root mean square deviation (RMSD), root mean square fluctuation (RMSF), radius of gyration (RG), solvent accessible surface area (SASA), and hydrogen bonding (H-Bond).

### 5.6. MM-PBSA Analysis

In order to confirm and affirm the accuracy of the docking score obtained from Autodock Vina, the ligand binding affinity determination based on The Molecular Mechanics Poisson–Boltzmann Surface Area (MM-PBSA) was performed. For this study, MMPBSA calculations were performed on the complex crystal structure of the EHMT2 SET domain with F80, N47, TP1, and TP2. The last 50 ns of Gromacs trajectories for each complex were used for the ΔG calculation. First, the complex structures were prepared for calculation using the Gromacs software-2019.4, including adding explicit solvent and generating topology files. The MMPBSA calculation was set up in the g MMPBSA software-2019.4, and the energy decomposition was performed on each complex over the last 50 ns of the trajectory. The resulting energy components were analyzed to determine the binding affinity and the contributions of different energy terms to the overall binding energy. The ΔG was calculated by using the following equation
ΔG_binding_ = G_complex_ − (G_protein_ + G_ligand_)(1)
where G_complex_ represents the free energy of protein–ligand complex (PL, where ligand here can either be substrate or inhibitor), G_protein_ and G_ligand_ denote the corresponding free energies of protein and ligand.

The free energy in bound and unbound form is calculated by
G_x_ = (E_MM_) − TS + (G_solv_)(2)
where X denotes PL complex or unbound form, namely P or L, the average molecular mechanics is calculated by E_MM_, TS refers to the entropic contribution, and G_solv_ corresponds to free energy solvation of ligand binding to protein.

The molecular mechanics (E_MM_) was determined by taking the electrostatic and Van der Waal’s (bonded and non-bonded) interactions between ligand and protein, as depicted in Equation (3). G_solv_ corresponds to the linear Poisson–Boltzmann equation for individual states (G_polar_), and solvent-accessible surface area was calculated using non-hydrophobic interactions.
E_MM_ = E_bonded_ + E_non-bonded_(3)
G_solv_ = G_nonpolar_ + G_polar_(4)

The results of the MMPBSA calculation were used to evaluate the binding of the ligands to the crystal structure of the EHMT2 SET domain enzyme and to identify potential binding sites and interactions.

## Figures and Tables

**Figure 1 molecules-28-07266-f001:**
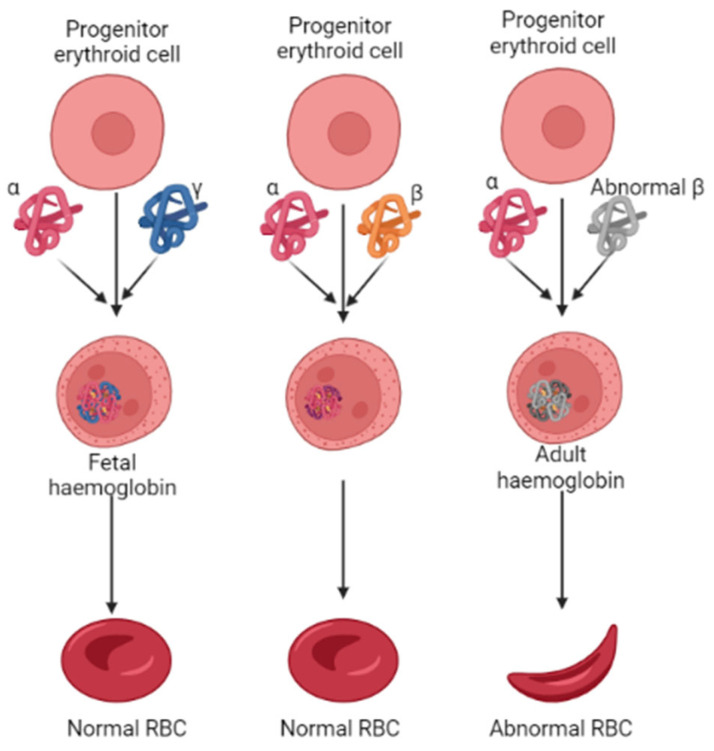
Pathological features of erythropoiesis in developing normal and abnormal RBC in β-thalassemia. The abnormal synthesis of β-globin levels characterizes β-thalassemia.

**Figure 2 molecules-28-07266-f002:**
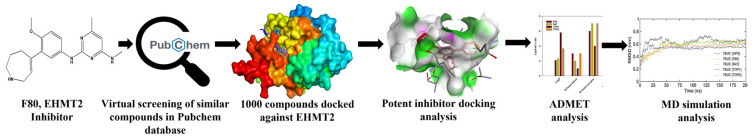
Summary of the proposed work demonstrating the discovery of novel potential EHMT2 inhibitors based on molecular docking and dynamic simulation studies.

**Figure 3 molecules-28-07266-f003:**
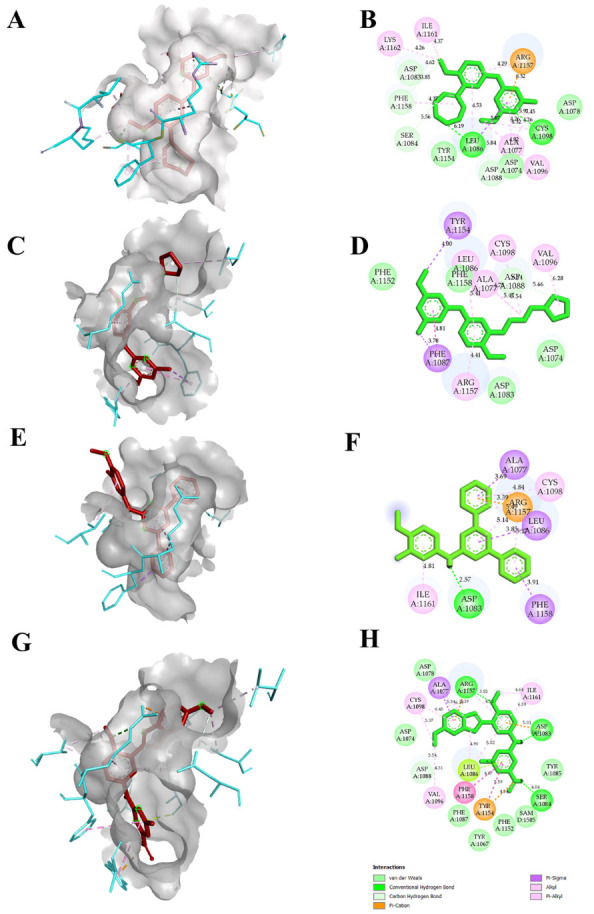
Molecular interactions of F80, N47, and other putative drug targets of β-thalassemia obtained through Autodock in 3D and 2D views. In (**A**,**C**,**E**,**G**) the red stick represents the docked ligand and the cyan color in the stick form represents the amino acids interacting with the docked ligand. The crystal structure corresponding to protein EHMT2 used for docking is 7BUC. (**A**,**B**) F80 interactions with EHMT2. (**C**,**D**) N47 interactions with EHMT2. (**E**,**F**) TP1 interactions with EHMT2. (**G**,**H**) TP2 interactions with EHMT2.

**Figure 4 molecules-28-07266-f004:**
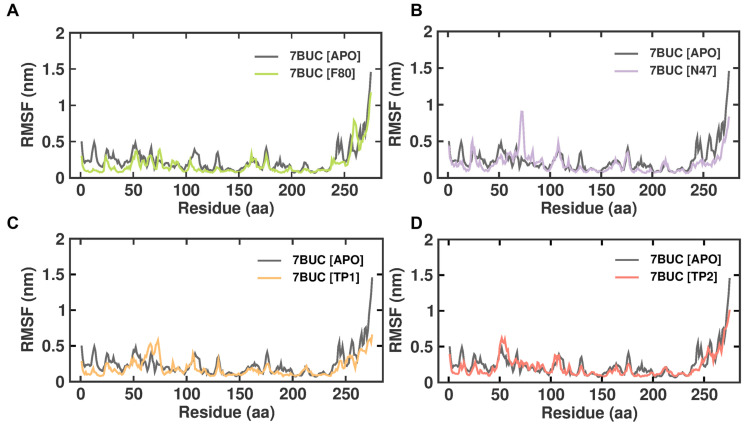
RMSD plot of docked complexes generated through MDS at 200 ns. (**A**). RMSD plot of 7BUC complexed with F80. (**B**) RMSD plot of 7BUC complexed with N47. (**C**) RMSD plot of 7BUC complexed with TP1, and (**D**) RMSD plot of 7BUC complexed with TP2.

**Figure 5 molecules-28-07266-f005:**
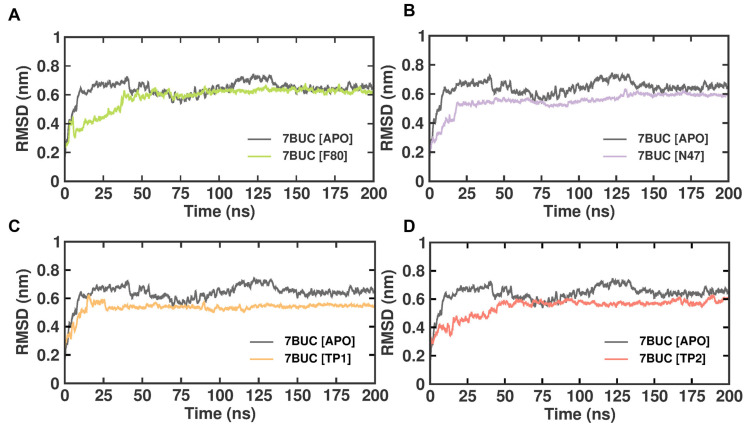
RMSF plot of docked complexes generated through MDS at 200 ns. (**A**). RMSF plot of 7BUC complexed with F80. (**B**) RMSF plot of 7BUC complexed with N47. (**C**) RMSF plot of 7BUC complexed with TP1, and (**D**) RMSF plot of 7BUC complexed with TP2.

**Figure 6 molecules-28-07266-f006:**
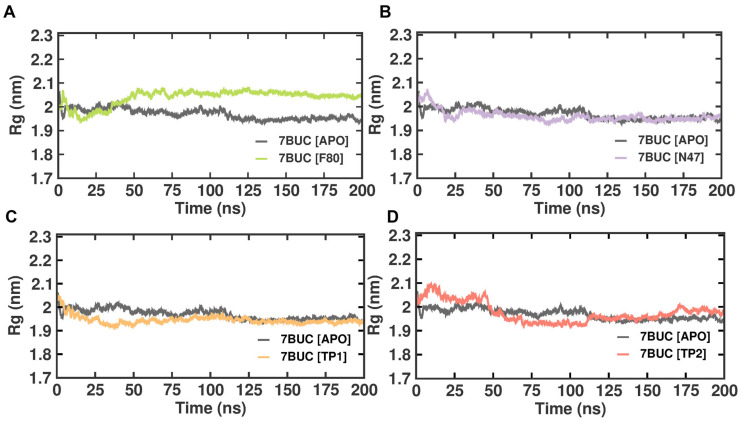
Rg plot of docked complexes generated through MDS at 200 ns. (**A**). Rg plot of 7BUC complexed with F80. (**B**) Rg plot of 7BUC complexed with N47. (**C**) Rg plot of 7BUC complexed with TP1, and (**D**) Rg plot of 7BUC complexed with TP2.

**Figure 7 molecules-28-07266-f007:**
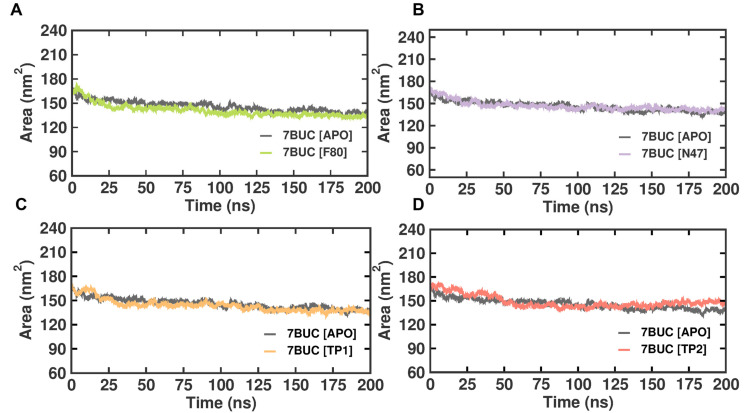
SASA plot of docked complexes generated through MDS at 200 ns. (**A**). SASA plot of 7BUC complexed with F80. (**B**) SASA plot of 7BUC complexed with N47. (**C**) SASA plot of 7BUC complexed with TP1, and (**D**) SASA plot of 7BUC complexed with TP2.

**Figure 8 molecules-28-07266-f008:**
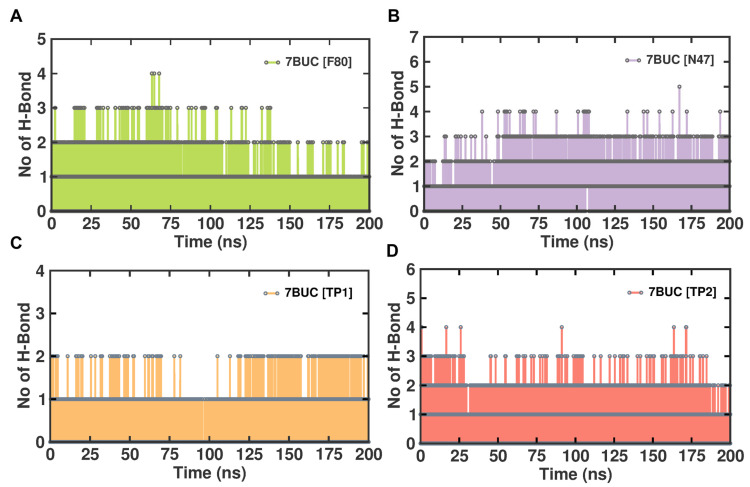
H-bond analysis of docked complexes generated through MDS at 200 ns. (**A**). Number of H-bond formed between 7BUC complexed with F80. (**B**) Number of H-bond formed between 7BUC complexed with N47. (**C**) Number of H-bonds formed between 7BUC complexed with TP1, and (**D**) Number of H-bonds formed between 7BUC complexed with TP2.

**Figure 9 molecules-28-07266-f009:**
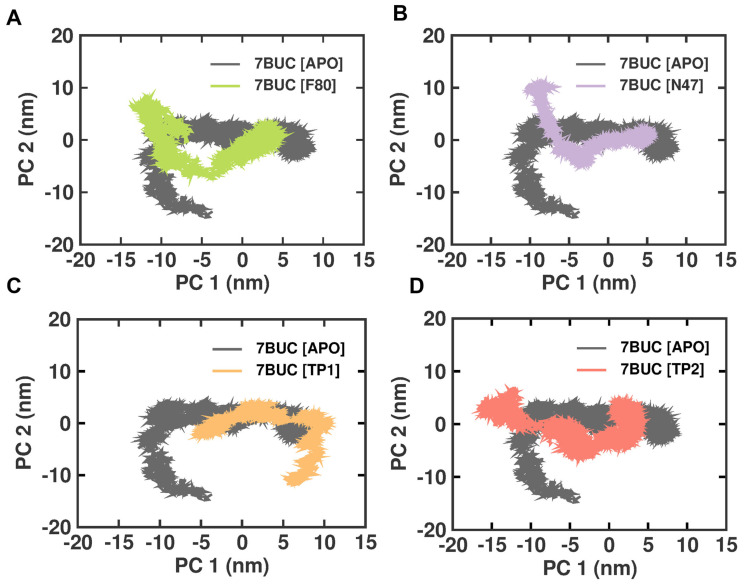
PCA analysis of docked complexes generated through MDS at 200 ns. (**A**). Clustering results formed between 7BUC complexed with F80. (**B**) Clustering results formed between 7BUC complexed with N47. (**C**) Clustering results formed between 7BUC complexed with TP1, and (**D**) Clustering results formed between 7BUC complexed with TP2.

**Table 1 molecules-28-07266-t001:** The compound name, PubChem ID, and top ten IFD scores obtained during docking studies.

Ligand	Code	PubChem ID	Autodock Score (Kcal/mol)
*N*-2-[4-Methoxy-3-(2,3,4,7-tetrahydro-1*H*-azepin-5-yl)phenyl]-*N*4,6-dimethyl-pyrimidine-2,4-diamine [Compound 13]	F80	154815704	8.2
*N*~2~-{4-Methoxy-3-[3-(pyrrolidin-1-yl)propoxy]phenyl}-*N*~4~,6-Dimethylpyrimidine-2,4-diamine [Compound 5]	N47	132021720	−7.5
*N*-(4-Methoxy-3-methylphenyl)-4,6-diphenylpyrimidin-2-amine	TP1	70841761	−10.7
2-*N*-[4-Methoxy-3-(5-methoxy-3*H*-indol-2-yl)phenyl]-4-*N*,6-dimethylpyrimidine-2,4-diamine	TP2	159271530	−10.3
*N*-(4-Methoxyphenyl)-6-phenyl-2-piperidin-4-ylpyrimidin-4-amine	TP3	135371993	−10.3
*N*-[4-(4-Methoxy-3-methylanilino)-6-phenylpyrimidin-2-yl]acetamide	TP4	70841736	−10.2
2-*N*-[(2-Methoxyphenyl)methyl]-6-methyl-4-*N*-[4-(trifluoromethyl)phenyl]pyrimidine-2,4-diamine	TP5	112918384	−10.1
2-*N*-[3-(3a,7a-dihydro-1*H*-pyrrolo [2,3-c]pyridin-2-yl)-4-methoxyphenyl]-4-*N*,6-dimethylpyrimidine-2,4-diamine	TP6	146397923	−9.9
2-*N*-[4-Methoxy-3-(2-methyl-3*H*-inden-5-yl)phenyl]-4-*N*,6-dimethylpyrimidine-2,4-diamine	TP7	159557168	−9.9
2-*N*-[(2-Methoxyphenyl)methyl]-6-methyl-4-*N*-(2,3,4-trifluorophenyl)pyrimidine-2,4-diamine	TP8	112918416	−9.8
2-*N*-(3,4-Difluorophenyl)-4-*N*-[(2-methoxyphenyl)methyl]-6-Methylpyrimidine-2,4-diamine	TP9	112918292	−9.8
2-*N*-[3-(1-Cyclobutyl-2,3,4,7-tetrahydroazepin-5-yl)-4-methoxyphenyl]-4-*N*,6-dimethylpyrimidine-2,4-diamine	TP10	162656337	−9.8

**Table 2 molecules-28-07266-t002:** Drug-likeness properties of the selected drug compounds obtained using Swiss-ADME.

Property	F80	N47	TP1	TP2
Molecular formula	C_19_H_25_N_5_O	C_20_H_29_N_5_O_2_	C_24_H_21_N_3_O	C_22_H_23_N_5_O_2_
Molecular weight (g/mol)	339.43	371.48	367.44	389.45
H-bond donors	3	2	1	2
H-bond acceptors	4	5	3	5
LogP (octanol/water partition coefficient)	2.831	3.132	6.244	4.037
Topological polar surface area	71.1	71.540	47.040	80.660
Lipinski’s rule of five	Passed	Passed	Passed	Passed
Ghose filter	Passed	Passed	No	Yes
Veber’s rule	Passed	Passed	Yes	Yes
Bioavailability score	0.55	0.55	0.55	0.55
Sp3	No	Passed	Passed	Passed
Golden Triangle	Passed	Passed	Passed	Passed

**Table 3 molecules-28-07266-t003:** The table presents the results of various energy calculations observed in four systems: 7BUC-F80, 7BUC-N47, 7BUC-TP1, and 7BUC-TP2.

Complex Name	Van der Waals Energy (kJ/mol)	Electrostatic Energy (kJ/mol)	Polar Solvation Energy (kJ/mol)	Binding Energy (kJ/mol)
7BUC-F80	−171 ± 10	−0.1 ± 5	69 ± 12	−119 ± 11
7BUC-N47	−224 ± 16	−28.5 ± 12	149 ± 35	−126 ± 22
7BUC-TP1	−230 ± 9	−26.1 ± 7	138 ± 22	−141 ± 17
7BUC-TP2	−178 ± 14	−544.1 ± 23	276 ± 24	−465 ± 24

## Data Availability

The data presented in this study are available on request from the corresponding author.

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
