# Peer review of "In Silico Molecular Docking and Dynamics Simulation Analysis of Potential Histone Lysine Methyl Transferase Inhibitors for Managing β-Thalassemia"

_molecules, 2023, doi:10.3390/molecules28217266_

Round 1

Reviewer 2 Report

The authors of the manuscript titled “In silico molecular docking and dynamics simulation analysis of potential histone lysine methyl transferase inhibitors obtained from PubChem database for managing β-thalassemia” applied similarity search and extensive virtual screening procedures to identify novel euchromatin histone lysine methyl transferase inhibitors. Molecular dynamics and free-binding energy calculations were performed.

The manuscript is a good piece of work, and the authors used many computational tools and analyses to identify the inhibitors.

My overall comment is accepted after minor corrections.

1-     The title, I suggest the authors change the title and remove the name of the database (PubChem) from the title and keep it in the abstract and introduction.

2-     I suggest the authors render methods number 2, after the introduction, then Results number 3. Because in Virtual screening, the readers need to know the procedures first before the results or discussion

3-     Figure 2, remove “Graphical abstract”  from the legend. Authors can upload the graphical abstract separately and will appear in front of the publication.

4-     Results, line 200 “2.2. Toxicity Analysis” I suggest replacing it with Drug likeness or physicochemical properties. As authors include information about drug likeness under this title, not only toxicity.

5-     The subheading under sector 2.3 Molecular Dynamics Simulations, needs to have numbers, not Bullets. For example, 2.3.1 RMSD analysis, 2.3.2 RMSF analysis and continue

6-     Line 276, RMSF is Figure 5, not Figure 6, correct, and review all the Figures number

7-     Discussion, very long. I suggest authors remove the part that is like the introduction about β-thalassemia and focus on discussing the results

8-     Figure 3, The ligands inside the binding pocket, are not clear. I suggest changing the  green colour to a more visible colour

9-     Figure S6, Authors need to mention which time in the simulation they generate the images

Round 2

Reviewer 1 Report

The authors have addressed all the points of the review report. The manuscript that results improved in clarity and soundness. I agree on the publication of the manuscript in the present form.